# Gestational weight gain at the national, regional, and income group levels based on 234 national household surveys from 70 low-income and middle-income countries

**Janaína Calu Costa**[1,2]\*, **Dongqing Wang**[3], **Molin Wang**[4,5], **Enju Liu**[6,7], **Uttara Partap**[1], **Ilana Cliffer**[1], **Wafaie W. Fawzi**[1,6,8]

1 Department of Global Health and Population, Harvard T.H. Chan School of Public Health, Boston, Massachusetts, United States of America, 2 International Center for Equity in Health, Federal University of Pelotas, Pelotas, Rio Grande do Sul, Brazil, 3 Department of Global and Community Health, College of Public Health, George Mason University, Fairfax, Virginia, United States of America, 4 Department of Epidemiology, Harvard T.H. Chan School of Public Health, Harvard University, Boston, Massachusetts, United States of America, 5 Department of Biostatistics, Harvard T.H. Chan School of Public Health, Harvard University, Boston, Massachusetts, United States of America, 6 Institutional Centers for Clinical and Translational Research, Boston Children's Hospital, Boston, Massachusetts, United States of America, 7 Division of Gastroenterology, Hepatology and Nutrition, Harvard Medical School, Boston Children's Hospital, Boston, Massachusetts, United States of America, 8 Department of Nutrition, Harvard T.H. Chan School of Public Health, Harvard University, Boston, Massachusetts, United States of America

\* jcosta@hsph.harvard.edu

**Data Availability Statement:** The datasets were anonymized to ensure confidentiality and are publicly available through the institutions' websites:

## Abstract

Gestational weight gain (GWG) estimates enable the identification of populations of women at risk for adverse outcomes. We described GWG distribution in low- and middle-income countries (LMICs). Demographic and Health Surveys and other national surveys were used to calculate the average GWG by regressing the weight of pregnant women (15–49 years) at the time of the interview on their gestational age, adjusting for sociodemographic factors. A mixed-effects hierarchical model was built with survey-specific GWG as the dependent variable and restricted cubic splines for survey year, super-region, and country-level covariates (total fertility rate, gross domestic product, and average female body mass index) to predict the national, regional, and income level average GWG in 2020. Uncertainty ranges (UR) were obtained using bootstrap. Estimates were compared with the Institute of Medicine's GWG recommendations for women with normal weight (11.5kg) and underweight (12.5kg). Survey data were available for 70 LMICs (234 data points, 1991–2022). Predicted country-specific GWG for 2020 ranged from 2.6 to 13.5kg. Ten countries presented estimates above the recommendation for women with underweight; nine of which were from Central Europe, Eastern Europe, and Central Asia; apart from one, these were upper-middle income. Regional GWG was estimated at 5.4kg (95%UR 3.1,7.7) in Sub-Saharan Africa; 6.2kg (95%UR 3.4,9.0) in North Africa and the Middle East; 8.6kg (95%UR 6.0,11.3) in South Asia; 9.3kg (95%UR 6.2,12.3) in Southeast Asia, East Asia, and Oceania; 10.0kg (95%UR 7.1,12.9) in Latin America and the Caribbean; and 13.0kg (95%UR 9.0,16.9) in Central and Eastern Europe, and Central Asia. A gradient was observed across income: 5.3kg (95%UR 2.7,7.9) for low-income, 7.6kg (95%UR 5.2,10.1) for lower-middle-income,

https://dhsprogram.com/data/, https://www.
datosabiertos.gob.pe/, https://bvsms.saude.gov.br/
bvs/pnds/, https://anda.inec.gob.ec/anda/index.
php/catalog/891, https://ensanut.insp.mx/
encuestas/ensanut2018/informes.php, https://
www.ine.gob.bo/index.php/encuesta-de-
demografia-y-salud-edsa-2016.

**Funding:** This research was supported by the Bill
and Melinda Gates Foundation (INV-016436 to
WWF). The funder had no role in the study design,
data collection and analysis, decision to publish, or
manuscript preparation. The authors had full
access to all the data in the study and had final
responsibility for the decision to submit for
publication.

**Competing interests:** The authors have declared
that no competing interests exist.

and 9.8kg (95%UR 7.1,12.5) for upper-middle-income countries. No income group achieved the minimum recommended weight gain. GWG was estimated to be insufficient in almost all LMICs. Improved data and monitoring are crucial for impactful interventions.

## Introduction

The amount of weight a woman gains during pregnancy, known as gestational weight gain (GWG), is important for the health of the pregnancy and the long-term well-being of the woman and the offspring [1]. It is part of the essential process for a woman's body to provide adequate support for the optimal growth and development of the fetus and is a potentially modifiable pregnancy factor as psychological, behavioral, family, social, cultural, and environmental aspects can have an impact on the weight gained during pregnancy [2, 3]. Insufficient maternal weight gain during pregnancy is associated with low birth weight, small- and short-for-gestational-age babies, and failure to initiate breastfeeding [4, 5]. On the other hand, excessive weight gain increases the risk of obstetric complications, cesarean deliveries, and post-partum weight retention for the mother and prematurity, large for gestational age newborns, macrosomia, and childhood overweight or obesity for the offspring [1, 4, 6].

To monitor trends of adequate weight gain during pregnancy over time and across countries, estimates of GWG at the national and regional levels, especially in low-resource settings, are crucial. Nevertheless, most of the available evidence on GWG is from high-income countries while in low- and middle-income countries (LMICs), little information is available. In these settings, due to the occurrence of both a high prevalence of undernutrition and an increasing prevalence of overweight and obesity, women starting pregnancy are vulnerable to a range of nutritional concerns [7].

Our previous study, by Wang et al. (2020), represented the first known effort to characterize and compare the average GWG levels across all LMICs. These analyses were based on a modeling approach of Demographic and Health Surveys (DHS), and a major burden of inadequate GWG in most LMICs and regions was reported [8]. In 2015, the mean GWG in Latin America and the Caribbean (11.8 kg) and Central Europe and Eastern Europe (11.2 kg) were substantially higher than the estimates for Sub-Saharan Africa (6.6 kg) and North Africa and the Middle East (6.8 kg); these latter, with estimates below 60% of the minimum recommendation defined by the Institute of Medicine (IOM) [8]. Also, all super-regions were below the minimum GWG for normal-weight women, as recommended by the IOM. Although the IOM guidelines were developed for pregnant women in the United States and are not intended for assessing other populations, they have been widely adopted by other countries [3, 9].

A different analytical approach using DHS data was developed for obtaining mean estimates as well as temporal trends in GWG among women from Sub-Saharan African countries [10]. For that, the authors used information from non-pregnant women of reproductive age considered at "risk of conception" (not abstaining from sex and not using any contraceptive methods) to estimate pre-pregnancy weight and subtract this from the estimated total weight at the end of the pregnancy. The total estimated GWG in the region was 6.6 kg, similar to the previously-mentioned work, and did not show meaningful changes over the previous 15 years [10].

In contrast, a study of 33 cohorts from Europe, North America, and Oceania participating in the LifeCycle Consortium, most high-income countries, found a median total GWG of a full-term pregnancy of 14.0 kg, with no differences in the patterns of weight gain between cohorts or countries [11]. The difference in GWG between countries at different national income levels suggests inequalities influenced by this characteristic.

Many individual risk factors for weight gain below or above the recommended amount have been described for LMICs, such as woman's body mass index (BMI), mid-upper arm circumference, height, smoking habit, and HIV infection [12]. In addition to these, disparities in social and economic factors may also play a role in determining whether pregnant individuals can achieve the recommended levels of weight gain [13–15]. GWG has been described at the population level by geographic regions, but to our knowledge, no work has also explored its distribution by countries' national income level.

Given limited longitudinal nationally representative data from LMICs, initiatives to use widely available surveys, such as DHS, to estimate and monitor weight gain during pregnancy have been developed as described above. With the present analysis, we aimed to use data from DHS and other national surveys to expand and update the previous analyses by describing GWG distribution in LMICs in the year 2020 and calculating these estimates by country income groups in addition to geographic regions.

## Methods

Our analyses included publicly available nationally representative household surveys conducted in LMICs. We used DHS and other national surveys with similar sampling designs that allow comparable indicators to be estimated. As of September 2023, a total of 234 household surveys were identified and met the criteria to be included in the analysis, including 12 national surveys not conducted by the DHS Program: eight of Peru's Demographic and Family Health Surveys [*Encuesta Demográfica y de Salud Familiar* (ENDES) 2013 to 2020] [16]; the 2006 Brazil's National Demographic and Health Survey (*Pesquisa Nacional de Demografia e Saúde*) [17]; the 2018 Ecuador's National Health and Nutrition Survey [*Encuesta Nacional de Salud y Nutrición* (ENSANUT)] [18]; the 2016 Bolivia's Demographic and Health Survey [*Encuesta de Demografía y Salud* (EDSA)] [19]; and the 2018 Mexico's National Health and Nutrition Survey [*Encuesta Nacional de Salud y Nutrición* (ENSANUT)] [20]. Non-DHS microdata have been pre-processed by the International Center for Equity in Health (Federal University of Pelotas, Brazil) in order to make them comparable to the standard surveys [21]. Ethical approval was the responsibility of the institutions in charge of each survey, and all of them provided anonymized microdata publicly; therefore, no ethical approval was required for the present work.

In all surveys, women of reproductive age (15–49 years) were selected by a two-stage stratified sampling process to respond to the interview [22]. In the first stage, enumeration areas (EA) were generally drawn from census files, while in the second stage, a sample of households was randomly selected from each EA. In each household, a questionnaire was completed to identify eligible members who were interviewed using an individual questionnaire. Each survey's Individual Record dataset was used, restricting the analytical sample to women who were pregnant at the time of the interview and for whom information on pregnancy duration and body weight was available.

We obtained country-specific average GWG in a given year by adapting the methodology developed by Coffey (2015), who originally employed it to DHS data from India and Sub-Saharan African countries collected around 2005 and was used by Wang et al (2020) to calculate global GWG levels for the year 2015 [8, 23]. For that, a hierarchical mixed-effects model was developed to use the estimated total weight gain from the surveys as the dependent variable and survey year and country-level covariates as the independent variables, enabling comparisons across geographies [8, 23].

To calculate weight gain in each survey, the individual cross-sectional body weight measure in kilograms (kg) was regressed on the gestational age in months using an ordinary least

squares model. As weight gain trajectories during the second and third trimesters are not expected to fail linearity, linear models were used. Only women in the second or third trimesters of pregnancy were included in the analysis, and the models accounted for the clustered nature of the data in each survey. Gestational age in months was calculated from complete days, weeks, or months based on the time since the last menstrual period (LMP); the information was supplemented by the self-reported duration of pregnancy in complete months whenever the time since the LMP was unavailable, based on the answer to the question "How many months pregnant are you?". As covariates, we included women's age and total years of education, the number of children ever born by the time of the interview, area of residence (rural or urban), and quintiles of household wealth. The survey-specific wealth index from which the quintiles were derived was constructed via principal components analysis using household asset data; the index is provided in most of the DHS databases, and the methodology was replicated in the non-DHS ones [24]. The survey-specific coefficients from the regression models (β) were interpreted as the average weight gain during the second and third trimesters of pregnancy in a given country and year.

Total GWG was used as the dependent variable in the hierarchical models and calculated by the following equation: $\beta * 6.5 * (1 + P)$, where β is the survey-specific coefficient indicating weight gain as described above; 6.5 indicates the average duration of the two last trimesters of a full-term pregnancy in months; and P is a relative percentage of weight gained in the third trimester in comparison to the second trimester. The latter information was obtained from GWG charts of the LifeCycle Consortium. More details on these steps are presented in supplementary material (S1 Appendix).

The hierarchical mixed-effects modeling approach was used to obtain the predicted estimates for 2020. The predictors included in the model were survey year and country-level covariates, the selection of which was guided by the model fit based on the lowest Bayesian Information Criteria (BIC) employed in the previous paper [8]. Models with only a linear term for year and with additional restricted cubic splines terms for year using different numbers (3, 4 or 5) and locations of knots were compared and the final model included restricted cubic splines for the survey year, geographical super-region, average adult female BMI (in kilograms per square meter), log-transformed gross domestic product (GDP) per capita, and total fertility rate (TFR, average number of births per woman).

The super-regions were defined according to the Global Burden of Diseases, Injuries, and Risk Factors Study (GBD) classification, based on epidemiological similarity and geographic closeness as follows: South-East Asia, East Asia, and Oceania; Sub-Saharan Africa; South Asia; Latin America and the Caribbean; North Africa and the Middle East; Central Europe, Eastern Europe and Central Asia. The average adult female's BMI was retrieved from NCD Risk Factor Collaboration (NCD-RisC) (23), and GDP and TFR were obtained from the World Bank data repository [25]. For this updated analysis, we used GDP and TFR for 2020 and mean female BMI for 2016, the latest estimates available.

In addition to the country-level estimates, average GWG was calculated for each of the six GBD super-regions and the corresponding regions: South-East Asia, East Asia and Oceania (South-East Asia, Oceania, East Asia); Sub-Saharan Africa (Western, Southern, Central, Eastern); South Asia; Latin America and Caribbean (Tropical, Caribbean, Andean, Central); North Africa and Middle East; Central Europe, Eastern Europe and Central Asia (Central Asia, Central Europe, Eastern Europe); and by country income level according to the World Bank classification for the year 2020 (low income, lower-middle income and upper-middle income) [26]. The estimates at the super-regional and regional levels and income-group level were derived from the weighted national estimates by the number of births in each country in 2020; the values were obtained from the United Nations World Population Prospects 2022 [27]. For all

point estimates, 95% uncertainty ranges (95% UR) were generated using non-parametric bootstrapping and a multiple imputation approach. This involves generating 10 pseudo data sets based on original point estimates and corresponding standard error (SE) estimates. Bootstrapping is applied to each pseudo data set to generate 10 pseudo point estimates and SE estimates. More information on the methods is available elsewhere [8].

For descriptive purposes, we compared the estimates with the IOM recommendations of weight gain during pregnancy: between 11.5 and 16 kg for women with normal weight (pre-pregnancy BMI 18.5–24.9 kg/m$^2$) and between 12.5 and 18 kg for women with underweight (pre-pregnancy BMI less than 18.5 kg/m$^2$) [9].

All analyses were conducted using Stata$^®$ version 17.0 (StataCorp. 2021. Stata Statistical Software: Release 17. College Station, TX: StataCorp LLC.) and SAS$^®$ OnDemand for Academics (SAS Institute Inc).

## Results

The final database comprised 234 surveys with weight data for pregnant women from 70 countries. Survey years ranged from 1991 to 2022 (median year = 2007), and the number of surveys available by country ranged from one to 16. The complete list of surveys and countries is available in the supplementary material (S1 Table). Countries with surveys included in this study represent 77.4% of all low-income, 59.6% of all lower-middle income, and 30.5% of all upper-middle-income countries in 2020. Regarding the super-regions, the percentage of included countries was 16% from Southeast Asia, East Asia, and Oceania; 33% from North Africa and the Middle East; 38% from Central Europe, Eastern Europe, and Central Asia; 48% from Latin America and the Caribbean; 80% from South Asia; and 80% from Sub-Saharan Africa.

The results from the hierarchical model indicate that at the national level, weight gain estimated for the year 2020 ranged from 2.6 kg in Central African Republic (95% UR -2.9, 8.2) and Samoa (95% UR -5.2,10.4) to 13.5 kg in Bosnia and Herzegovina (95% UR -10.3, 16.7). Country-specific estimates and the corresponding 95% uncertainty ranges are presented in the supplementary material (S2 Table). Only ten countries presented point estimates above the lower limit of IOM-recommended GWG for women with underweight (12.5 kg), and nine of them were from Central Europe, Eastern Europe, and Central Asia super-region (Montenegro, Belarus, Romania, Ukraine, North Macedonia, Russia, Serbia, Bulgaria, Bosnia and Herzegovina); the tenth country was from the Latin America and Caribbean super-region (Brazil). Except for Ukraine, all these countries were categorized as upper-middle income geographies in the year 2020. The country-specific estimates are presented in Figs 1 and 2 by super-region and income group, respectively.

The country-specific analysis was extended to encompass six global super-regions and corresponding regions (Table 1). The estimated values were 5.4 kg (95% UR 3.1, 7.7) in Sub-Saharan Africa; 6.2 kg (95% UR 3.4, 9.0) in North Africa and Middle East; 8.6 kg (95% UR 6.0, 11.3) in South Asia; 9.3 kg (95% UR 6.2, 12.3) in Southeast Asia, East Asia, and Oceania; 10.0 kg (95% UR 7.1, 12.9) in Latin America and Caribbean; and 13.0 kg (95% UR 9.0, 16.9) in Central Europe, Eastern Europe, and Central Asia.

Only two out of the three Central Europe, Eastern Europe, and Central Asia regions achieved the IOM recommendations for normal-weight women, with the third (Central Asia) presenting a borderline estimate of 11.4 kg (95% UR 7.8, 15.0). On the other hand, the lowest regional estimates were observed in Central Sub-Saharan Africa 4.4 kg (95% UR 2.0, 6.9), Eastern Sub-Saharan Africa 5.7 kg (95% UR 3.4, 8.1), and Oceania 5.8 kg (95% UR 2.1, 9.5).

We also estimated the average GWG by income group and a gradient was observed, with increasing weight gain as higher the per capita income: 5.3 kg (95% UR 2.7, 7.9) for low-

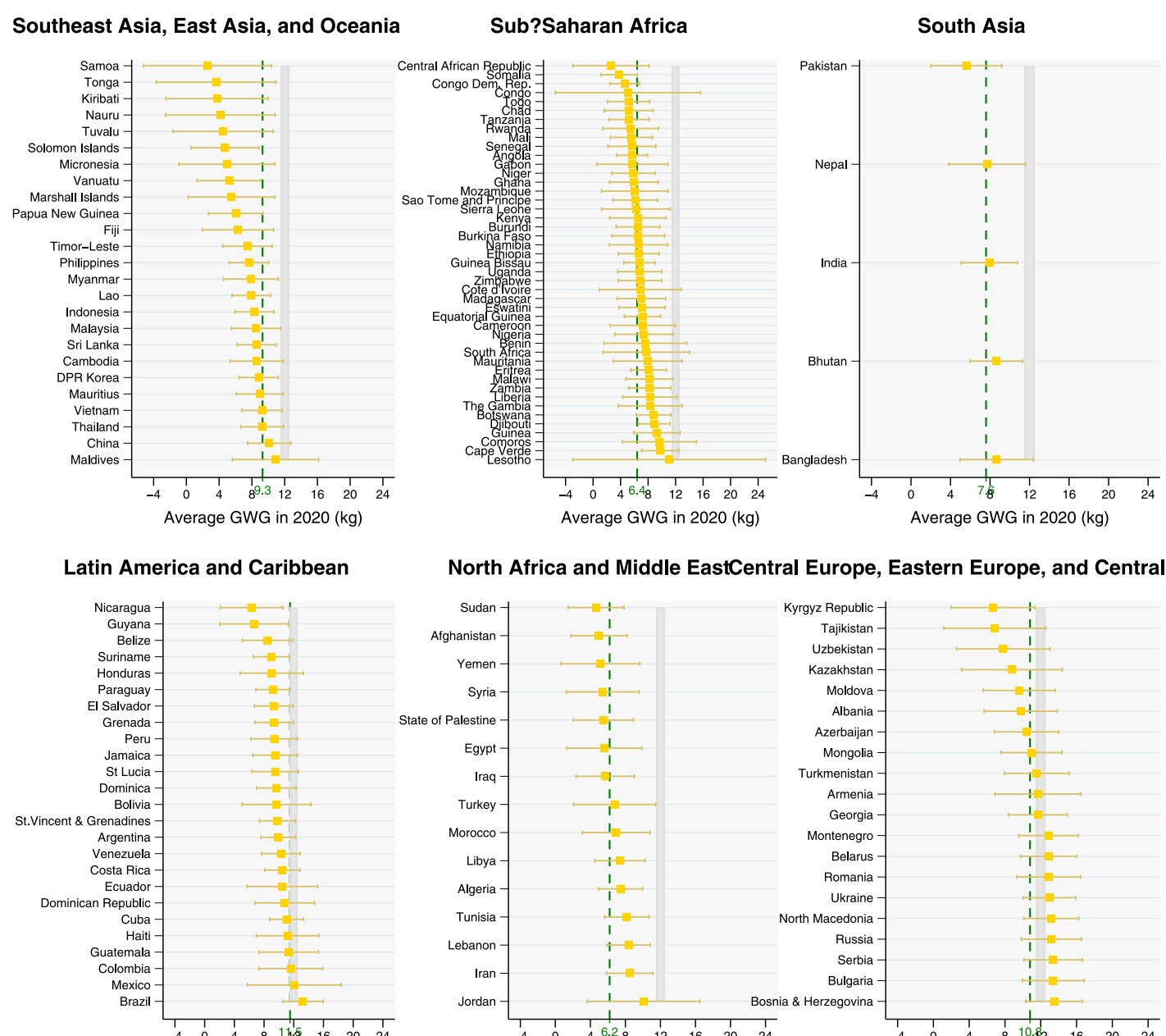

**Fig 1. Average country-specific gestational weight gain (GWG) estimated for the year 2020 using hierarchical modeling and the corresponding 95% uncertainty range by geographic super-region.** Note: dashed lines represent the super region estimate. The gray area represents the minimum total gestational weight gain recommended by the Institute of Medicine for normal-weight (11.5 kg) and underweight (12.5 kg) women.

income, 7.6 kg (95% UR 5.2, 10.1) for lower-middle-income, and 9.8 kg (95% UR 7.1, 12.5) for upper-middle-income countries (Table 1). None of the groups achieved the IOM recommended weight gain for women with either underweight or normal weight.

## Discussion

Using hierarchical modeling of cross-sectional data from 234 national surveys from 70 countries and nationally representative data from publicly available repositories, this study

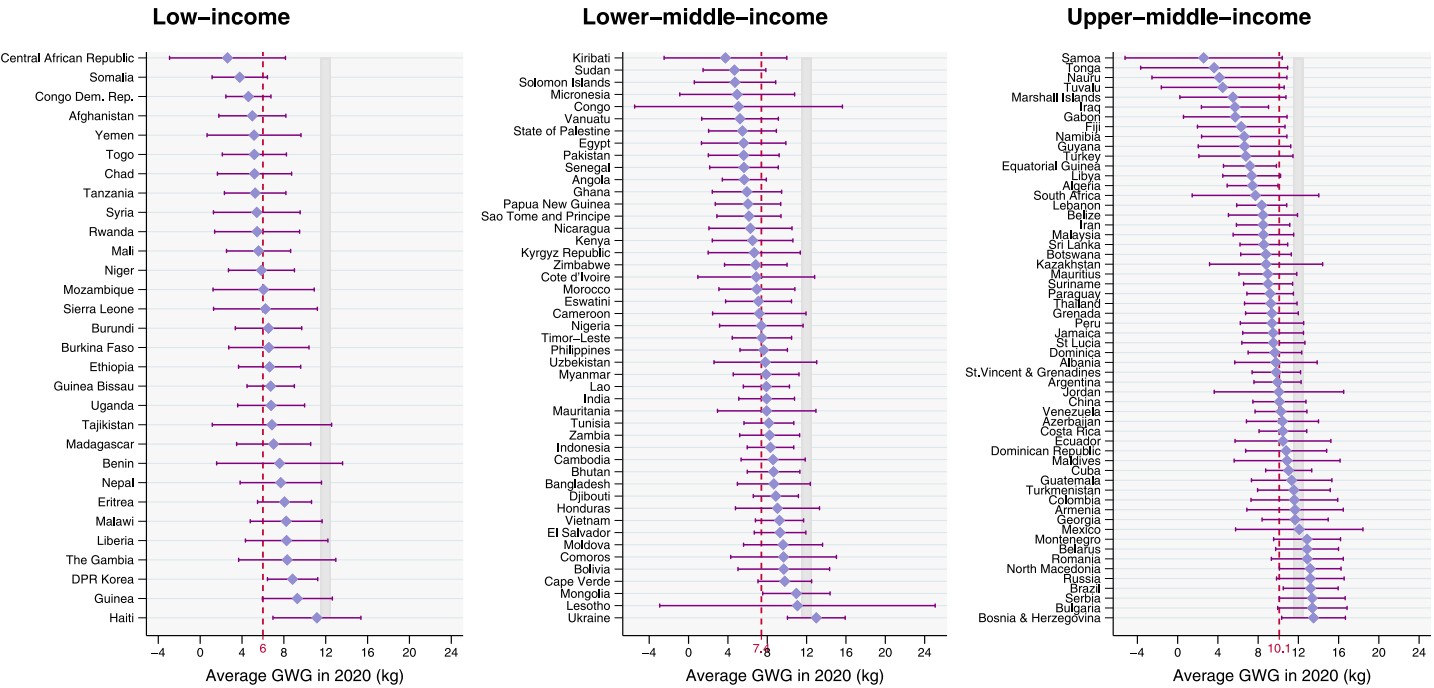

**Fig 2. Average country-specific gestational weight gain (GWG) estimated for the year 2020 using hierarchical modeling and the corresponding 95% uncertainty range by income group.** Note: dashed lines represent the income group estimate. The gray area represents the minimum total gestational weight gain recommended by the Institute of Medicine for normal-weight (11.5 kg) and underweight (12.5 kg) women.

estimated the weight gain during pregnancy in the year 2020 for the group of LMICs at the national, regional, and income group levels. Our analysis showed that, in most countries, the amount of gestational weight gained by women is, on average, below the IOM recommendations, with only those in Central Europe, Eastern Europe, and Central Asia region presenting an average GWG within the expected range for both women with normal weight and underweight. In sub-Saharan Africa, North Africa and the Middle East, and South Asia, there were observed the lowest estimates, with GWG below 9 kg, which represents about less than 80% of the IOM minimum recommendations of 11.5 kg for normal weight and 12.5 kg for women with underweight; in some countries and regions less than 50% of the IOM recommendation has been achieved. Our results reinforce the evidence of stark disparities between LMICs and high-income geographies, as an average GWG of 14.0 kg was observed for a group of countries from Europe, North America, and Oceania [11].

We also observed a positive gradient in the average weight gain during pregnancy across country income groups, wherein the low-income group presented the lowest GWG estimate, and upper-middle-income countries presented the highest value. As described in other works, this gradient was also observed for maternal undernutrition, as assessed by different outcomes, with a higher prevalence among low-income countries and the poorest groups in middle-income countries [7]. This finding underscores the complex interplay between socioeconomic factors and maternal health outcomes, as socioeconomic inequalities are known as substantial and persistent determinants of undernutrition worldwide [7]. Globally, 10% of women aged 20–49 years (around 154 million) suffer from underweight, with poorer regions and countries presenting a disproportionate share of these [28]. In LMICs, in particular, where resources are often constrained and access to healthcare services may be limited, maternal undernutrition is more prevalent as a consequence of multiple contributing factors such as poverty, food

**Table 1. Gestational weight gain (GWG) estimates for the year 2020 by regions and national income level derived from hierarchical modeling.**

| Group | Number of Countries | GWG Estimate (kg) | 95% Uncertainty Range | |
|---|---|---|---|---|
| Region | | | Lower Bound | Upper Bound |
| **Central Europe, Eastern Europe, and Central Asia** | **21** | **13.0** | **9.0** | **16.9** |
| Central Asia | 9 | 11.4 | 7.8 | 15.0 |
| Central Europe | 8 | 13.1 | 9.2 | 17.0 |
| Eastern Europe | 4 | 13.1 | 9.6 | 16.7 |
| **Latin America and the Caribbean** | **25** | **10.0** | **7.1** | **12.9** |
| Caribbean | 3 | 10.4 | 7.4 | 13.4 |
| Central Latin America | 11 | 10.1 | 7.2 | 13.1 |
| Tropical Latin America | 8 | 9.8 | 7.3 | 12.3 |
| **North Africa and the Middle East** | **3** | **6.2** | **3.4** | **9.0** |
| **South Asia** | **15** | **8.6** | **6.0** | **11.3** |
| **Southeast Asia, East Asia, and Oceania** | **5** | **9.3** | **6.2** | **12.3** |
| East Asia | 25 | 10.1 | 7.1 | 13.0 |
| Oceania | 2 | 5.8 | 2.1 | 9.5 |
| Southeast Asia | 11 | 8.4 | 5.6 | 11.2 |
| **Sub-Saharan Africa** | **12** | **5.4** | **3.1** | **7.7** |
| Central Sub-Saharan Africa | 46 | 5.7 | 3.4 | 8.1 |
| Eastern Sub-Saharan Africa | 6 | 4.4 | 2.0 | 6.9 |
| Southern Sub-Saharan Africa | 15 | 8.8 | 6.3 | 11.3 |
| Western Sub-Saharan Africa | 6 | 7.2 | 4.7 | 9.6 |
| **Country Income Level** | | | | |
| Low-income | 31 | 5.3 | 2.7 | 7.9 |
| Lower-middle-income | 47 | 7.6 | 5.2 | 10.1 |
| Upper-middle-income | 59 | 9.8 | 7.1 | 12.5 |

insecurity, and inadequate healthcare infrastructure. These lead women to enter pregnancy with compromised nutritional status, and they consequently face heightened risks of insufficient GWG. These findings are of concern since it is well-documented that insufficient weight gain is associated with several adverse pregnancy outcomes [4]. Furthermore, if a woman experiences inadequate or excessive GWG, it may set the stage for a cycle where her offspring are at an increased risk of facing similar nutritional challenges, potentially leading to a repeating pattern of malnutrition across generations [7, 29]. Therefore, maintaining optimal GWG and preventing insufficient and excessive weight gain during pregnancy is essential to promote maternal well-being and ensure optimal fetal development.

To our knowledge, only a few other studies used nationally representative survey data to estimate GWG in LMICs. In India and Sub-Saharan Africa, women were estimated to gain around 7 kg during a full-term pregnancy [23]. Consistently, another study estimated GWG in Sub-Saharan Africa was estimated at 6.6 kg and did not find changes over time [10].

Despite not presenting the average weight gain in kg, evidence from individual patient data meta-analyses of longitudinal studies from LMICs indicated a burden of suboptimal GWG as assessed using the IOM recommendations. Among 55 primary studies conducted in different regions (mainly from sub-Saharan Africa, Latin America and the Caribbean, and South Asia), the prevalence of inadequate GWG was 53.9%, and excessive GWG was observed among 22% of the participants [4,12]. Among these, inadequate GWG ranged from 16% in a study from China to 88% in a study from Nepal; regarding excessive GWG, the prevalence ranged from 2% in a study from Malawi to 58% in a study from Pakistan [12].

Although there have been important reductions in the prevalence of underweight in LMICs, progress has been uneven across and within regions and countries; at the same time, the prevalence of overweight and obesity among women has increased over the decades in these countries [7, 30, 31]. It has been described that more than a third of LMICs presented such overlapping forms of malnutrition with shifts in its occurrence towards lower-income countries from the 1990s to 2010s [31]. A study conducted in a city in southern Brazil—a middle-income country—compared maternal anthropometrics in four population-based birth cohorts and found that the prevalence of overweight or obesity at the beginning of the pregnancy increased from 22.1% to 47.0% between 1982 and 2015 [32]. This observed change in the epidemiological profile of LMICs may be accompanied by an increase in the occurrence of excessive GWG, as observed in high-income countries. Continued research is needed to determine which risk factors for inadequate and excessive GWG are most relevant to the design of effective public health interventions promoting healthy weight gain in these settings.

At the individual level, it has also been described that women in lower socioeconomic positions tend to gain less weight during pregnancy [10, 15]. Nevertheless, studies addressing inequalities in GWG at the individual level are mostly from high-income countries where concerns about overweight and obesity, as well as excessive weight gain during pregnancy, are more common [33]. In some of them, higher socioeconomic position was found not to provide a protective effect against excessive GWG among overweight and obese women [34].

The study findings underscore the importance of public health interventions for maternal nutrition. Comprehensive measures are required to tackle the underlying factors contributing to suboptimal GWG—including both inadequate and excessive weight gain during pregnancy —in these settings, including proper nutrition at the beginning of the gestation and ensuring healthy pregnancies that reduce the likelihood of adverse outcomes [30]. Global agencies have established guidelines and policy priorities to address such global concerns. For instance, the United Nations Children's Fund (UNICEF) has developed a strategic framework encompassing five priorities to achieve the objectives associated with the prevention of all forms of malnutrition in women, including programming for women's nutrition before and during pregnancy and women's nutrition during the breastfeeding period, nutrition of adolescent mothers and other nutritionally at-risk women, and innovations for maternal nutrition [30]. Under this framework, several interventions targeted at meeting the specific needs of malnourished women are highlighted, including social protection, screening and treatment for anemia, micronutrient deficiencies, helminth infections, and undernutrition [30]. The 2016 "WHO Recommendations on Antenatal Care for a Positive Pregnancy Experience" document also reinforces the importance of nutrition interventions during pregnancy and guides the practice, organization, and delivery of antenatal care services [35]. In light of the findings of this study, maternal nutrition interventions and monitoring should always apply an equity lens as an integral component toward optimal outcomes at the population level [36]. Public health nutritional interventions such as targeted balanced energy and protein supplementation for pregnant women in low-income and food-insecure contexts, as well as provision of micronutrient supplementation and nutrition counseling, have been recognized as crucial for changing the health trajectories of insufficient improvement on women's health during pregnancy [7, 37]. Our results contribute to these initiatives by providing evidence to inform the designing of interventions to address GWG in LMICs, reinforcing the need for tailored GWG monitoring tools for the LMICs' context and community-based programs to mitigate the burden of suboptimal GWG and other associated morbidities.

This study, however, has some limitations. First, since we are using multiple cross-sectional surveys, we were unable to estimate GWG longitudinally, and the results are based on aggregated estimates rather than individual-level estimates. Small country-specific and regional

differences in both directions were observed between estimates for the year 2015, as described by Wang et al (2020), and those resulting from the present analysis for the year 2020, which are likely to be a result of the increased number of surveys in the model, and no changes over time can be implied [8]. As we observed, the proportion of countries with available surveys ranged from 16% to 80% by super-region; therefore, their representativeness can be limited. Also, information on pregnancy status and duration relies on maternal reporting and recall and thus is more likely to be biased than objective measurements. No pre-pregnancy information is available, preventing us from estimating weight gain for different pre-pregnancy BMI; also, given data constraints we were not able to estimate GWG by individual characteristics or the contribution of other determinants even at the population level. From the modeling perspective, current estimates for this set of countries are subject to change due to the addition of new data. We are also aware that the estimates of GWG presented in this work might mask diversity at the subnational level, particularly in highly populated and less homogeneous countries. Finally, the GWG recommendations for women with normal weight and underweight are based on IOM guidelines and might not accurately reflect what would be expected for pregnant women from diverse populations from LMICs as it was developed based on data primarily from North America. Even though efforts have been made to develop international standards and references for GWG monitoring, no global tool is yet available, and the IOM guidelines are still the most widely used tool for this purpose; therefore, caution is important when interpreting these findings, as the generalizability may be limited.

Despite the limitations, findings from this study add to the emerging body of literature in low- and middle-income settings, suggesting the poorer situation of these countries regarding GWG adequacy. In addition to efforts to properly monitor weight gain during pregnancy in LMICs through nationally representative studies with a longitudinal design, using survey data can be an effective tool to monitor trends of GWG distributions in order to develop and implement interventions. National household surveys constitute the main data source for nutrition indicators in LMICs, and our work expands a previously published analysis by adding nationally representative surveys conducted in LMICs beyond the DHS scope and updating the analytical database by including new surveys that have become available in the past years. The previous work included 206 DHS (1991–2018) from 67 countries [8]. This updated analysis included two new countries (Ecuador and Mexico) and 28 new surveys not previously analyzed, 12 of which were not standard DHS. Finally, we incorporated income level analysis, and the findings reinforce this evidence of a socioeconomic gradient in GWG. Analyses such as those can motivate commitment toward focusing on the countries and regions where intervention is most needed.

Ending malnutrition in all its forms (including undernutrition, micronutrient-related malnutrition, overweight, obesity, and resulting related noncommunicable diseases) is among the top priorities of the United Nations Decade of Action on Nutrition (2016–2025) and is part of the 2030 Agenda for Sustainable Development [38, 39]. Suboptimal GWG is a strong predictor of poor pregnancy outcomes and has been identified in most LMICs with marked inequalities across regions and income groups; however, systematic monitoring of weight gain during pregnancy or assessment of pre-pregnancy BMI at the population level is rarely done in these settings. Our work strengthens the call for high-quality, timely, and reliable data on pregnancy weight gain in LMICs to better identify the levels, trends, and determinants of GWG. Encouraging further research to understand and intervene in GWG trends in specific countries and regions is also important. This research will aid in crafting interventions that suit specific factors influencing weight gain during pregnancy, leading to more impactful intervention strategies.

## Supporting information

**S1 Appendix. Description of the analytical framework.**
(DOCX)

**S1 Table. List of surveys (country and year) included in the analysis and the corresponding super-region, region, and income group.**
(DOCX)

**S2 Table. Country-specific gestational weight gain (GWG) estimates for the year 2020 derived from hierarchical modeling and the corresponding 95% uncertainty range limits.**
(DOCX)

## Acknowledgments

We thank the International Center for Equity in Health team (Federal University of Pelotas, Brazil, https://equidade.org/) for sharing pre-processed and standardized survey microdata used in the analyses.

## Author Contributions

**Conceptualization:** Janaína Calu Costa, Dongqing Wang, Molin Wang, Wafaie W. Fawzi.

**Data curation:** Janaína Calu Costa.

**Formal analysis:** Janaína Calu Costa, Dongqing Wang.

**Funding acquisition:** Wafaie W. Fawzi.

**Methodology:** Janaína Calu Costa, Dongqing Wang, Enju Liu.

**Project administration:** Uttara Partap, Wafaie W. Fawzi.

**Supervision:** Dongqing Wang, Molin Wang, Enju Liu, Wafaie W. Fawzi.

**Visualization:** Janaína Calu Costa.

**Writing – original draft:** Janaína Calu Costa, Dongqing Wang, Uttara Partap.

**Writing – review & editing:** Janaína Calu Costa, Dongqing Wang, Molin Wang, Enju Liu, Uttara Partap, Ilana Cliffer, Wafaie W. Fawzi.

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
