## [Decision Letter · Decision Letter 0]

24 May 2024

PGPH-D-24-00785

Gestational weight gain at the national, regional, and income group levels based on 234 national household surveys from 70 low-income and middle-income countries

Dear Dr. Costa,

Thank you for submitting your manuscript to PLOS Global Public Health. After careful consideration, we feel that it has merit but does not fully meet PLOS Global Public Health’s publication criteria as it currently stands. Therefore, we invite you to submit a revised version of the manuscript that addresses the points raised during the review process.

Normal gestational weight gain (GWG) is multifactorial and varies in each country, with pre-pregnancy body mass index (BMI) being a stronger predictor. Hence, authors are advised to be cautious about applying the Institute of Medicine (IOM) recommendations universally to all pregnant women, especially in low- and middle-income countries (LMICs), where women tend to have a lower weight. The IOM guidelines were created for North American women and were not intended to be used for other populations. As of now, most GWG charts published were primarily derived from populations in high-income countries (HICs) or do not cover the entire range of pre-pregnancy body mass index (BMI).  The IOM guidelines and other GWG charts have limited generalizability, and do not apply to most low- and middle-income countries (LMICs). The authors have the opportunity to discuss this limitation and advise countries to develop their reference regional or country-specific guidelines or propose that local standards be developed, considering the rising prevalence of obesity in LMICs which heightens the risk of gestational diabetes and hypertensive disorders. Authors should discuss the implications of optimal GWG, emphasizing the importance of avoiding both excessive and insufficient weight gain.

In most LMICs, there is a growing burden of gestational weight gain, GDM and gestational hypertension, especially when women have higher pre gestational BMI. Authors can recommend the development of standardized GWG guidelines tailored to local contexts and the implementation of community-based monitoring programs  to mitigate the burden of gestational weight gain (GWG), GDM, and gestational hypertension in LMICs.

Given the positive response from reviewers, I am likely to accept your article without further peer review, provided that you resolve the minor comments from the reviewers and myself.

We look forward to receiving your revised manuscript.

Kind regards,

Giridhara R Babu, MBBS, MPH, PhD

Academic Editor

Journal Requirements:

3. We ask that a manuscript source file is provided at Revision. Please upload your manuscript file as a .doc, .docx, .rtf or .tex.

4. Please provide separate figure files in .tif or .eps format only and remove any figures embedded in your manuscript file. Please also ensure all files are under our size limit of 10MB.

Additional Editor Comments (if provided):

Reviewers' comments:

Reviewer's Responses to Questions

**Comments to the Author**

1. Does this manuscript meet PLOS Global Public Health’s publication criteria? Is the manuscript technically sound, and do the data support the conclusions? The manuscript must describe methodologically and ethically rigorous research with conclusions that are appropriately drawn based on the data presented.

Reviewer #1: Yes

Reviewer #2: Yes

Reviewer #3: Yes

2. Has the statistical analysis been performed appropriately and rigorously?

Reviewer #1: Yes

Reviewer #2: Yes

Reviewer #3: I don't know

3. Have the authors made all data underlying the findings in their manuscript fully available (please refer to the Data Availability Statement at the start of the manuscript PDF file)?

Reviewer #1: Yes

Reviewer #2: Yes

Reviewer #3: Yes

4. Is the manuscript presented in an intelligible fashion and written in standard English?

Reviewer #1: Yes

Reviewer #2: Yes

Reviewer #3: Yes

5. Review Comments to the Author

Reviewer #1: Dear authors, your study is very interesting and has excellent clinical implications.

I have just a few comments.

1- Reference no. 2 could be expanded with more interesting studies

2- Line 76- lacks a bibliographic reference

3- Line 102- this is the first time the acronym BMI has appeared... it should initially be described in full.

4- IOM- the first time the acronym appears it should be in full.

5- Did you take the pregnant woman's pre-pregnancy weight into account for the average weight gain values? Women with pre-gestational obesity have a lower ranking for gestational weight gain. They are only considered normative weight and underweight, but as they mention in the discussion, there is an increasing number of obese women.

6- On the other hand, the study could have analyzed women of normal weight and low weight separately.

7- It would have been interesting to have the birth weight values of the newborns.

8- Why did they include surveys from the same country in different years if they didn't then do a comparative analysis of the evolution between the years?

9- It would be interesting to see if there has been a downward trend or a gain in gestational weight in recent years? You have a considerable percentage of surveys over 20 years old. Does it make sense to analyze them as a whole? A lot could have changed in 20 years...

10- Table 1- first row/second column. Is it the number of countries? I don't understand these numbers

Reviewer #2: Thank you for submitting this sophisticated analysis. Some points to note are listed below:

• At the end of the third paragraph of page 5 – can you clarify if you mean countries with different income groups or income groups within countries?

• At the end of page 5, there is a statement that no work has been done. However, there has been some recent work as per below:

o Darling et al. (2023) https://journals.plos.org/plosmedicine/article?id=10.1371/journal.pmed.1004236

o Wang et al. (2020)

https://gh.bmj.com/content/5/11/e003423.abstract

• For the discussion, a more extended summary of how the results compare with similar research on low- and middle-income countries would be good.

Reviewer #3: The authors have done extensive statistical analysis on household surveys from 70 countries to yield gestational weight gain estimates. These data are essential to public health planning and support of healthy pregnanices in these countries. The additional comparison of GWG to income levels demonstrates a principle that is widely known from experience-women in poorer areas gain less weight than those in higher income areas. The authors have fully acknowledged the limitations in their methods: maternal recall, few prepregnancy weights, and the estimations. The authors are congratulated in this significant effort that can be used worldwide.

6. PLOS authors have the option to publish the peer review history of their article (what does this mean?). If published, this will include your full peer review and any attached files.

**Do you want your identity to be public for this peer review?** For information about this choice, including consent withdrawal, please see our Privacy Policy.

Reviewer #1: No

Reviewer #2: No

Reviewer #3: No

---

## [Decision Letter · Decision Letter 1]

2 Aug 2024

Gestational weight gain at the national, regional, and income group levels based on 234 national household surveys from 70 low-income and middle-income countries

PGPH-D-24-00785R1

Dear Dr Costa,

We are pleased to inform you that your manuscript 'Gestational weight gain at the national, regional, and income group levels based on 234 national household surveys from 70 low-income and middle-income countries' has been provisionally accepted for publication in PLOS Global Public Health.

Best regards,

Giridhara R Babu, MBBS, MPH, PhD

Academic Editor

Reviewer Comments (if any, and for reference):

Reviewer's Responses to Questions

**Comments to the Author**

1. If the authors have adequately addressed your comments raised in a previous round of review and you feel that this manuscript is now acceptable for publication, you may indicate that here to bypass the “Comments to the Author” section, enter your conflict of interest statement in the “Confidential to Editor” section, and submit your "Accept" recommendation.

Reviewer #1: All comments have been addressed

Reviewer #3: All comments have been addressed

2. Does this manuscript meet PLOS Global Public Health’s publication criteria? Is the manuscript technically sound, and do the data support the conclusions? The manuscript must describe methodologically and ethically rigorous research with conclusions that are appropriately drawn based on the data presented.

Reviewer #1: Yes

Reviewer #3: Yes

3. Has the statistical analysis been performed appropriately and rigorously?

Reviewer #1: Yes

Reviewer #3: I don't know

4. Have the authors made all data underlying the findings in their manuscript fully available (please refer to the Data Availability Statement at the start of the manuscript PDF file)?

Reviewer #1: Yes

Reviewer #3: Yes

5. Is the manuscript presented in an intelligible fashion and written in standard English?

Reviewer #1: Yes

Reviewer #3: Yes

6. Review Comments to the Author

Reviewer #1: (No Response)

Reviewer #3: The results published have many potential limitations: country level surveys, maternal recall, no prepregnancy BMI so estimates are used and intra-country differences in socio-economic opportunities. In spite of these limitations, the authors present regional data that will be very useful in public health planning.

7. PLOS authors have the option to publish the peer review history of their article (what does this mean?). If published, this will include your full peer review and any attached files.

**Do you want your identity to be public for this peer review?** For information about this choice, including consent withdrawal, please see our Privacy Policy.

Reviewer #1: No

Reviewer #3: No
